# An Engineered λ Phage Enables Enhanced and Strain-Specific Killing of Enterohemorrhagic *Escherichia coli*

Menglu Jin,[a,b] Jingchao Chen,[b] Xueyang Zhao,[a,b] Guoru Hu,[b] Hailei Wang,[a] Zhi Liu,[c] Wei-Hua Chen[a,b,d]

[a]College of Life Science, Henan Normal University, Xinxiang, Henan, China

[b]Key Laboratory of Molecular Biophysics of the Ministry of Education, Hubei Key Laboratory of Bioinformatics and Molecular Imaging, Center for Artificial Intelligence Biology, Department of Bioinformatics and Systems Biology, College of Life Science and Technology, Huazhong University of Science and Technology, Wuhan, Hubei, China

[c]Department of Biotechnology, College of Life Science and Technology, Huazhong University of Science and Technology, Wuhan, China

[d]Institution of Medical Artificial Intelligence, Binzhou Medical University, Yantai, China

Menglu Jin, Jingchao Chen, and Xueyang Zhao contributed equally to this article. Author order was determined by the corresponding authors after negotiation.

**ABSTRACT** Bacteriophages (phages) are ideal alternatives to traditional antimicrobial agents in a world where antimicrobial resistance (AMR) is emerging and spreading at an unprecedented speed. In addition, due to their narrow host ranges, phages are also ideal tools to modulate the gut microbiota in which alterations of specific bacterial strains underlie human diseases, while dysbiosis caused by broad-spectrum antibiotics can be harmful. Here, we engineered a lambda phage (Eλ) to target enterohemorrhagic *Escherichia coli* (EHEC) that causes a severe, sometimes lethal intestinal infection in humans. We enhanced the killing ability of the Eλ phage by incorporating a CRISPR-Cas3 system into the wild-type λ (wtλ) and the specificity by introducing multiple EHEC-targeting CRISPR spacers while knocking out the lytic gene *cro*. *In vitro* experiments showed that the Eλ suppressed the growth of EHEC up to 18 h compared with only 6 h with the wtλ; at the multiplicity of infection (MOI) of 10, the Eλ killed the EHEC cells with ∼100% efficiency and did not affect the growth of other laboratory- and human-gut isolated *E. coli* strains. In addition, the EHEC cells did not develop resistance to the Eλ. Mouse experiments further confirmed the enhanced and strain-specific killing of the Eλ to EHEC, while the overall mouse gut microbiota was not disturbed. Our methods can be used to target other genes that are responsible for antibiotic resistance genes and/or human toxins, engineer other phages, and support *in vivo* application of the engineered phages.

**IMPORTANCE** Pathogenic strains of *Escherichia coli* are responsible for 0.8 million deaths per year and together ranked the first among all pathogenic species. Here, we obtained, for the first time, an engineered phage, Eλ, that could specifically and efficiently eliminate EHEC, one of the most common and often lethal pathogens that can spread from person to person. We verified the superior performance of the Eλ over the wild-type phage with *in vitro* and *in vivo* experiments and showed that the Eλ could suppress EHEC growth to nondetectable levels, fully rescue the EHEC-infected mice, and rescore disturbed mouse gut microbiota. Our results also indicated that the EHEC did not develop resistance to the Eλ, which has been the biggest challenge in phage therapy. We believe our methods can be used to target other pathogenic strains of *E. coli* and support *in vivo* application of the engineered phages.

**KEYWORDS** enterohemorrhagic *Escherichia coli*, EHEC, sequence-specific antimicrobial agent, strain-specific bacterium killing, engineered phage, CRISPR-Cas3

Address correspondence to Wei-Hua Chen, weihuachen@hust.edu.cn, Zhi Liu, zhiliu@hust.edu.cn, or Hailei Wang, whl@htu.cn.

The authors declare no conflict of interest.

Bacterial infection can seriously affect human health and even cause human death. In fact, the World Health Organization (WHO) has estimated that by 2050, bacterial infection can kill as many as 10 million people per year, more than cancer (1). Strikingly, *Escherichia coli*,

one of the most prevalent bacteria in human gut (2–5), was responsible for more than 0.8 million deaths in 2019 alone and ranked the first among all pathogens (6). Many *E. coli* strains or serotypes exist; most of them are commensal to their hosts, and some can even be beneficial (7, 8). Pathogenic strains of *E. coli* include enteropathogenic *E. coli* (EPEC), enterohemorrhagic *E. coli* (EHEC), enterotoxigenic *E. coli* (ETEC), enteroaggregative *E. coli* (EAEC), and enteroinvasive *E. coli* (EIEC) (9), among which, EHEC, one of the most common foodborne pathogens, can be transmitted to humans by food, drink, animal and environmental contact, or directly from person to person (8). Pathogens can be killed by antibiotics; however, antimicrobial resistance (AMR) has been emerging and spreading at an unprecedented speed, partly due to antibiotic abuse. In fact, most bacterial infection-related deaths are related to AMRs (6). In addition, most antibiotics can kill a broad spectrum of microbes, greatly disturb the intestinal microbiota, and lead to dysbiosis, which has been associated with infections by opportunistic bacteria (10), increased disease risks (11, 12), and decreased efficacy of many drugs (13). Therefore, there is an urgent need to specifically kill pathogenic *E. coli* strains such as EHEC without affecting others in the same community.

In recent years, bacteriophages (or phages for short) have been (re)recognized as an ideal alternative for traditional broad-spectrum antibiotics because of their high diversity in all biomes and host specificity. For example, it has been estimated that up to 56% of the phages have hosts at species and genus levels (14, 15). Phage therapy has been used to treat burn wounds, urogenital tract infections, respiratory tract infections, chronic otitis, and *E. coli* diarrhea recently (16). In addition, phages have been used to remove pathogens from foods (17, 18) and crops (19). However, limitations exist in the use of phages as an antimicrobial agent (AMA), including low efficiency in killing the target bacteria and quickly developing resistance by the host (20). To enhance the killing efficacy, researchers have developed many strategies, including the use of multiple phages (known as cocktails) for the same targets and engineered phages carrying host-targeting CRISPR spacers or even whole CRISPR-Cas systems (20–24). The latter strategy, also known as "sequence-specific" killing, uses the sequences incorporated as CRISPR spacers to guide the specificity of the targets and thus can be adapted to kill either a particular strain by using the strain-specific sequences as the spacers or a species by using the conserved sequences across its multiple strains. In addition, multiple spacers can be incorporated into the same engineered phage to target different sequences such as AMR and toxin genes.

Many CRISPR-Cas systems have been used for sequence-specific bacterial killing, especially CRISPR-Cas9 and CRISPR-Cas3. They were designed to target AMR genes or pathogenic bacteria genome(s) and were delivered into bacteria by packaging them into phages to achieve sequence-specific bacterial killing. For example, Bikard et al. used the ΦNM1 phage encoded with CRISPR-Cas9 to target antibiotic resistance of *Staphylococcus aureus* (25), while the research by Selle et al. shows that CRISPR-Cas3 can be delivered by ΦCD24-2 phages and specifically kill *Clostridioides difficile* in a sequence-specific manner via targeting of the bacterial genome (24, 26). Compared to Cas9, Cas3 is more efficient in terms of genome-scale deletions, such as the targeted removal of entire genes, gene clusters, islands, prophages, or plasmids (26–28).

In this study, we engineered a lambda phage to target specifically an EHEC strain, ATCC 35150. The engineered $\lambda$ phage (E$\lambda$) contains a CRISPR-Cas3 system and an EHEC-targeting CRISPR spacer. To increase the specificity of the E$\lambda$, we knocked out its lytic gene, *cro*, so that it used only the CRISPR-Cas3 system to kill its intended targets. *In vitro* experiments confirmed enhanced specificity and killing efficiency compared with the wild-type $\lambda$ phage (wt$\lambda$). We further showed in mouse experiments that E$\lambda$ could better protect the infected mice than the wt$\lambda$. Our methods can be used to engineer other phages, kill other pathogenic *E. coli* strains, and support *in vivo* application of the engineered phages.

## RESULTS

**An engineered phage with enhanced efficiency and specificity against EHEC.** To enhance the killing efficiency of the lambda phage ($\lambda$) against pathogenic enterohemorrhagic *Escherichia coli* (EHEC) strains, we engineered the $\lambda_{19014–27480}$ region of the wild-type

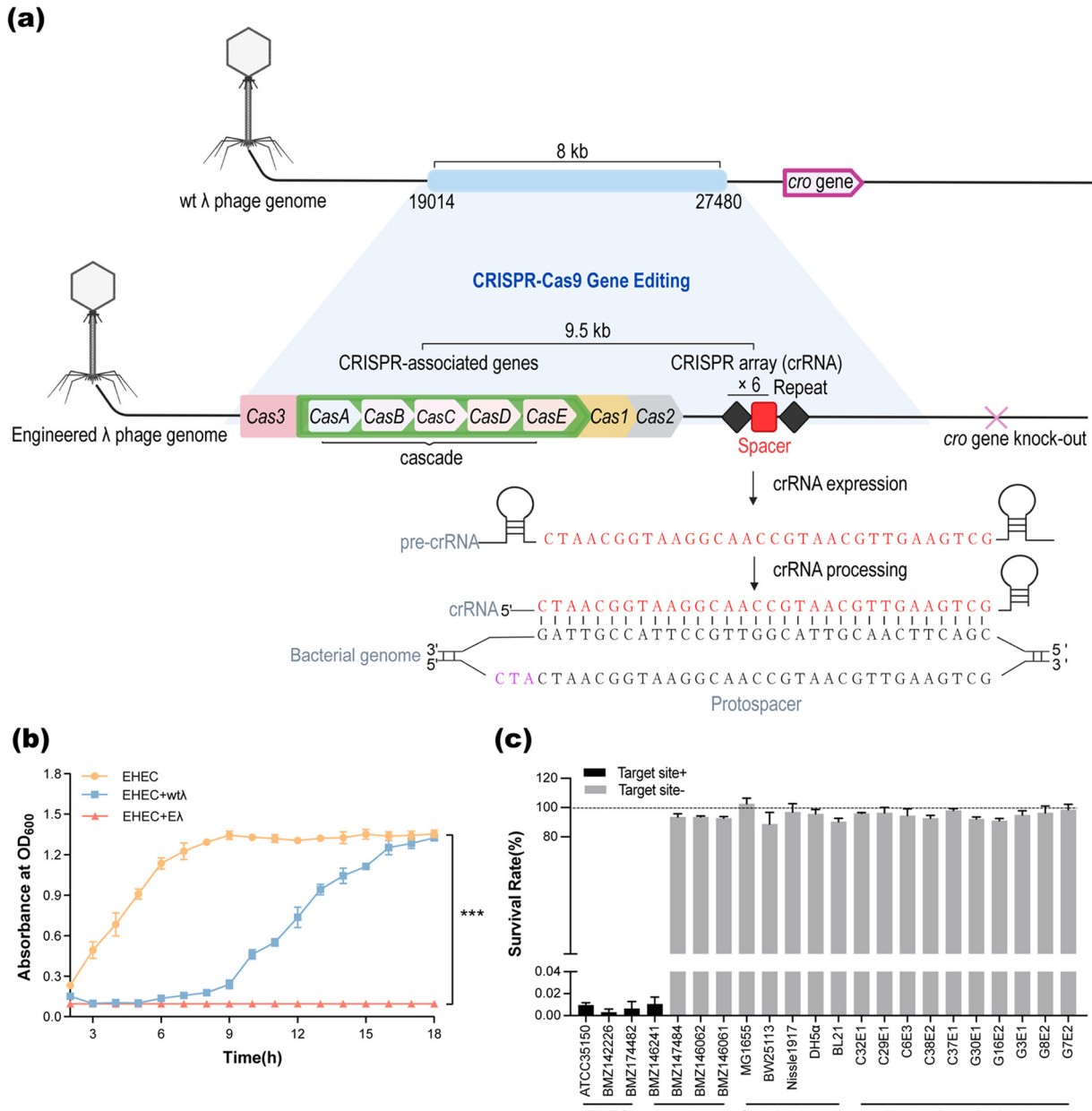

**FIG 1** An engineered λ phage with enhanced and strain-specific killing ability against enterohemorrhagic *Escherichia coli* (EHEC). (a) Overview of the engineering of the lambda (λ) phage. First, an 8-kb fragment on the wild-type λ phage (wtλ) was engineered to contain the *cas3* gene, the cascade genes, and a CRISPR array containing a spacer specifically targeting the *eae* gene of the *E. coli* EHEC strain; the spacer was repeated six times. In addition, the *cro* gene was knocked out from the wtλ genome. See Materials and Methods for more details. (b) Antibacterial curves of the wild-type (wtλ) and engineered (Eλ) phages against EHEC *in vitro*. (c) Killing specificity test of the Eλ against 22 *E. coli* strains, including 3 EHEC strains, 4 EPEC strains, 5 common laboratory strains, and 10 gut commensal strains isolated from human feces. "Target site+" indicates strains that contain the *eae* gene and can be targeted by the CRISPR spacer, while "Target site−" indicates strains that either do not contain the *eae* gene or the sequences of their *eae* genes do not match the CRISPR spacer. The survival rate was measured at the 12th hour after infection with an MOI (multiplicity of infection) of 10 and calculated as CFU (with Eλ)/CFU (without Eλ). All of the data are expressed as the mean ± SD. ($n = 3$; ***, $P < 0.001$, one-way ANOVA and Tukey's posttest).

λ phage (wtλ) that did not contain any known genes (23) to carry the CRISPR-Cas3-related genes and a CRISPR array (Fig. 1a; Materials and Methods). The CRISPR-Cas3 related genes were inserted and arranged into two operons; one contained the *casABCDE* and *cas3* genes and was controlled by the tac promoter; the other contained the *cas1* and *cas2* genes and was controlled by the J23119 (spel) promoter (Materials and Methods). The CRISPR array was also under the control of the J23119 (spel) promoter and contained a spacer targeting the *eae* gene of the *E. coli* EHEC strain (Materials and Methods); the spacer was repeated

six times in the array to increase the killing efficiency (Fig. 1a). The *eae* is a virulence gene located in the locus of enterocyte effacement (LEE) island and is essential for the pathogenicity of EHEC. In fact, among the 170 EHEC strains we have surveyed in the NCBI Prokaryotic Genome Database (see Table S1 in the supplemental material), 165 (97%) contained the *eae* gene (Materials and Methods; Fig. S1), suggesting that broad killing ability against the EHEC strains could be achieved by targeting the *eae* gene. We carried out the above engineering steps using a CRISPR-Cas9-mediated gene editing system in the *E. coli* AB2013329 strain in which the $\lambda$ phage existed as a prophage. In the end, the engineered region was expanded from 8 kb in size to 9.5 kb (Fig. S2).

To increase the killing specificity of the engineered $\lambda$ phage (E$\lambda$), we further knocked out its *cro* gene by using the CRISPR-Cas9-mediated gene editing system (Materials and Methods). The *cro* gene is an essential lytic regulator for phage entry into the lytic pathway (29–31).

We validated the engineered phage using PCR (Fig. S2) and obtained E$\lambda$ virions by induction with mitomycin (Materials and Methods). We further validated the ability of the E$\lambda$ to form plaques on the *E. coli* EHEC lawn, although the size of plaques was smaller than those of the wt$\lambda$ (Fig. S3).

We validated the killing efficiency and specificity of the E$\lambda$ against EHEC (ATCC 35150) by using *in vitro* experiments. First, we suppressed the growth of EHEC by using both the wt$\lambda$ and E$\lambda$ phages, and we found that the E$\lambda$ had significantly better ability to suppress the growth of EHEC than wt$\lambda$ at multiple multiplicities of infection (MOIs) ranging from 0.1 to 10 (Fig. S4a and Fig. 1b). Especially at an MOI of 10, we observed no EHEC growth up to 18 h after infection in the E$\lambda$ group, compared to only ~6 h of growth suppression by the wt$\lambda$ (Fig. 1b). At the MOI of 1, the EHEC treated by the E$\lambda$ resumed growth after ~8 h (Fig. 1b); to check whether this was due to resistance to the phage, we reinoculated the recovered EHECs in the LB medium and cocultured with additional E$\lambda$ to a final MOI of 10. We found that the EHEC growth was again completely suppressed, suggesting that the EHEC did not develop resistance to E$\lambda$ (Fig. S4c).

We then tested the killing specificity of wt$\lambda$ and E$\lambda$ using an additional 22 *E. coli* strains, including 3 EHEC strains, 4 enteropathogenic *E. coli* (EPEC) strains, 5 commonly used laboratory strains, and 10 gut commensal strains isolated from human feces in our laboratory (Fig. S4d and Fig. 1c). We calculated a survival rate for each of the strains by dividing their CFU in the E$\lambda$ treatment growth by those in the nontreatment group. We observed almost no survival of the three EHEC strains that all contained the *eae* gene (Table S2) at the 12th hour after infection of the E$\lambda$ with an MOI of 10 in contrast to ~100% of survival rate of the laboratory and gut commensal strains (Fig. 1c). Conversely, the wt$\lambda$ had a broad spectrum of killing against most of the 22 strains (Fig. S4d), but with much lower killing efficiency (the survival rates ranged from 59% to 100% with an average ~80% at the 12th hour after infection with an MOI of 10; Fig. S4d). These results confirmed the enhanced efficiency and specificity of E$\lambda$ against EHEC compared with the wt$\lambda$. Interestingly, one out of the four EPEC strains could also be effectively eliminated by the E$\lambda$, which contained the *eae* gene (the BMZ146241 strain) (Fig. 1c; Table S2), suggesting the E$\lambda$ could also be used to eliminate other *E. coli* strains containing the pathogenic *eae* gene.

**E$\lambda$ rescued EHEC-infected mice.** We further validated the effect of E$\lambda$ *in vivo* using an EHEC (ATCC 35150)-colonized mouse model. The experiment design is shown in Fig. 2a. Briefly, 40 mice were randomly divided into 4 groups, namely, the control, EHEC, EHEC plus wt$\lambda$, and EHEC plus E$\lambda$ groups. After acclimation and a food and water fasting period of 12 h, all mice were intraperitoneally injected with mitomycin to allow better EHEC infection (Materials and Methods). All mice except those in the control group were infected with EHEC 100 $\mu$L ($10^{10}$ CFU/mL) by gavage, followed by gavage with 100 $\mu$L of 10% sodium bicarbonate solution that could protect phage particles from damage by gastric acid. Then, the mice received one of three different treatments, phosphate-buffered saline (PBS) buffer, 100 $\mu$L wt$\lambda$ ($10^{10}$ PFU/mL), or 100 $\mu$L E$\lambda$ ($10^{10}$ PFU/mL) (Fig. 2a), according to their groups.

We observed a continuous decrease in body weight 2 days after the EHEC infection (day 3 of the experiment) (Fig. 2b) in the EHEC group compared with the control; the

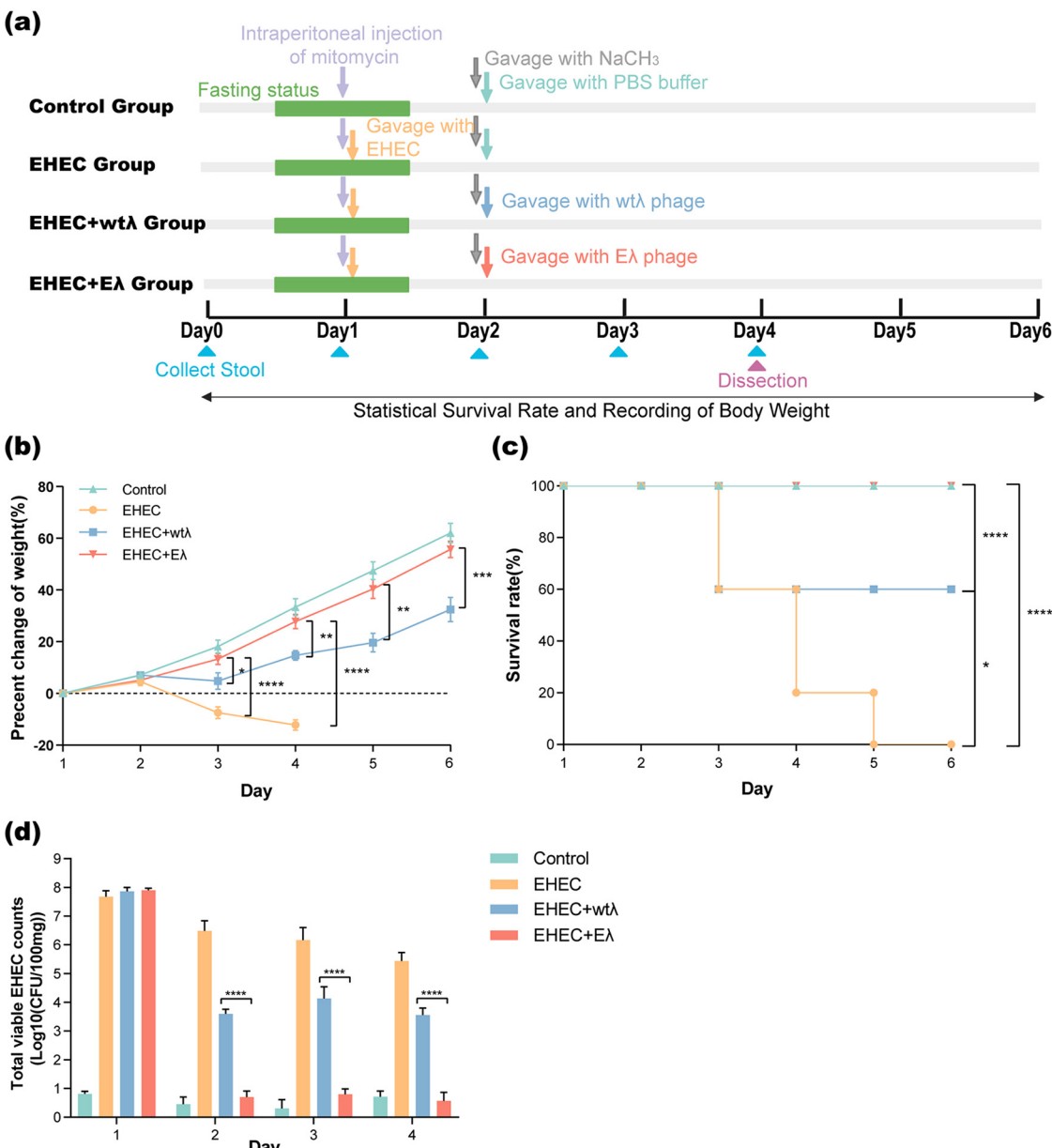

**FIG 2** Eλ eliminates EHEC *in vivo* and rescues EHEC-infected mice. (a) Experimental design using an EHEC infection mouse model. Ten mice were used for each of the four groups. (b) Weight changes (*y* axis) as a function of time (days, *x* axis) in the four groups. Weight changes were calculated as percent change from baseline weight on day 0 throughout the experiment. All data are expressed as the mean ± SD. (*n* = 10; *, $P < 0.05$; **, $P < 0.01$; ***, $P < 0.001$; ****, $P < 0.0001$; one-way ANOVA and Tukey's posttest). (c) Mouse survival rate (*y* axis) as a function of time (days, *x* axis) in the four groups. (*n* = 10; *, $P < 0.05$; ****, $P < 0.0001$; log-rank [Mantel-Cox] test). (d) Total viable EHEC counts (CFU) in 100 mg mouse feces. All data are expressed as mean ± SD (*n* = 10; ****, $P < 0.0001$, one-way ANOVA and Tukey's posttest).

differences were statistically significant ($P < 0.05$, one-way analysis of variance [ANOVA] and Tukey's posttest) (Fig. 2b). Mouse weight in the EHEC plus wtλ group (or wtλ group for short) also decreased significantly on day 3, but increased steadily from day 4, although it remained significantly lower than the control group from day 3 to the end of the experiment (Fig. 2b). Conversely, for mice in the EHEC plus Eλ group (or Eλ group for short), the weight was only slightly lower than that in the control group, with no significant difference (Fig. 2b). The mice in the EHEC and wtλ groups began to die on day 3, and the mortality rates reached to 100% and 40%, respectively, at the end of the experiment (Fig. 2c). These results indicate that the Eλ could fully rescue the mice from EHEC infection, while wtλ had limited protection against EHEC.

We also monitored the *in vivo* EHEC loads in the mouse feces of the four groups from day 1 to day 4 (Fig. 2a). Compared with the EHEC group, the EHEC loads in the wt$\lambda$ and E$\lambda$ groups were significantly reduced (Fig. 2d), suggesting that both could kill EHEC *in vivo*. However, we found that the EHEC strain was almost undetectable after the second day of infection, in contrast to $10^4$ CFU per 100 mg feces of the wt$\lambda$ group, further confirming the much higher killing efficiency of the engineered phage (Fig. 2d).

**E$\lambda$ alleviated EHEC-induced tissue damage and intestinal inflammation.** To evaluate the physiological conditions of the mice in different experimental groups, we dissected the mice on day 4 and collected the feces, blood, colon, liver, and kidney for further analysis. We found that the EHEC infection significantly shortened the colon lengths, which could be fully rescued by the E$\lambda$ phage treatment (Fig. 3a and Fig. S4). The wt$\lambda$ treatment also alleviated the EHEC-induced colon shortening, although no significant difference could be found compared with any of the other groups ($P > 0.05$, one-way ANOVA and Tukey's posttest). Furthermore, we examined tissue damage in colon, liver, and kidney using H&E staining (Fig. 3b and Fig. S5b; Methods) and found significantly higher inflammatory infiltration and goblet cell loss in the colon of the EHEC group (Fig. 3b). In addition, we found a significant increase of tissue damage in both liver and kidney (Fig. S5b). In comparison, these pathological features almost disappeared in the E$\lambda$ group (Fig. 3b and Fig. S5).

To quantify the above pathological features, we scored each of the edema, inflammation, and epithelial damage for the colon on a scale of 0 to 4 according to the evaluation criteria published by Selle et al. (24), and we found the highest total score in the EHEC group as we expected. The scores were significantly decreased in the E$\lambda$ group ($P < 0.05$, one-way ANOVA and Tukey's posttest), although we still found significantly higher scores in the E$\lambda$ group than the control group (Fig. 3c). The latter was likely caused by the fact that the E$\lambda$ was added 1 day later after the EHEC infection (Fig. 2a), and by then, certain damage had already been done to the mice.

We also examined the levels of selected inflammatory markers in the mouse serum and feces. We found significantly higher levels of proinflammatory factors, including interleukin 6 (IL-6), IgG2a, and IgG1 in the serum and IgA in the feces, in the EHEC group than in both the control and E$\lambda$ groups (Fig. 3). The E$\lambda$ treatment decreased all the markers to levels that were comparable to the control group ($P > 0.05$, one-way ANOVA and Tukey's posttest), indicating that the E$\lambda$-treated mice fully recovered from the EHEC infection. The wt$\lambda$ treatment also decreased all markers except the serum IgA, but certain markers, such as IL-6 and IGg2a, remained significantly higher than both the control and E$\lambda$ groups (Fig. 3e and g), suggesting the wt$\lambda$ was less efficient than the E$\lambda$, consistent with the *in vitro* experiments.

**E$\lambda$ alleviated EHEC-induced intestinal microbiota disruption.** EHEC infection is known to disrupt gut microbiota (32). We thus also checked if the phage treatments could alleviate the dysbiosis. We submitted the collected fecal samples (the V3 and V4 regions) on day 4 for 16S sequencing ($n = 4$ for each group; Materials and Methods). We compared the overall microbial community structures between groups measured by richness (Chao1 and ACE) and diversity (Shannon, Simpson) (33, 34), and observed the lowest Chao1, ACE, Shannon, and Simpson indexes (alpha diversity measurements) in the EHEC group. In addition, the indexes of the EHEC group were significantly different from those of the E$\lambda$ group ($n = 4$; *, $P < 0.05$; Wilcoxon rank-sum test). These indexes measure the richness and diversity of gut microbiota; lower index values suggested that EHEC infection indeed disrupted the gut microbiota. The E$\lambda$ treatment improved the mouse microbiota to a level similar to the control group (Fig. 4a). The wt$\lambda$ treatment also increased the alpha diversity of the mouse microbiota but was less efficient than the E$\lambda$ phage. These results also suggest that the E$\lambda$ phage did not significantly affect the gut microbiota when used *in vivo*. We further confirmed these observations by using principal-coordinate analysis (PCoA), which measured between-sample differences. As shown in Fig. 4b, the E$\lambda$ group was the closest to the control group, while samples in the EHEC group appeared to be in a state of disorder and

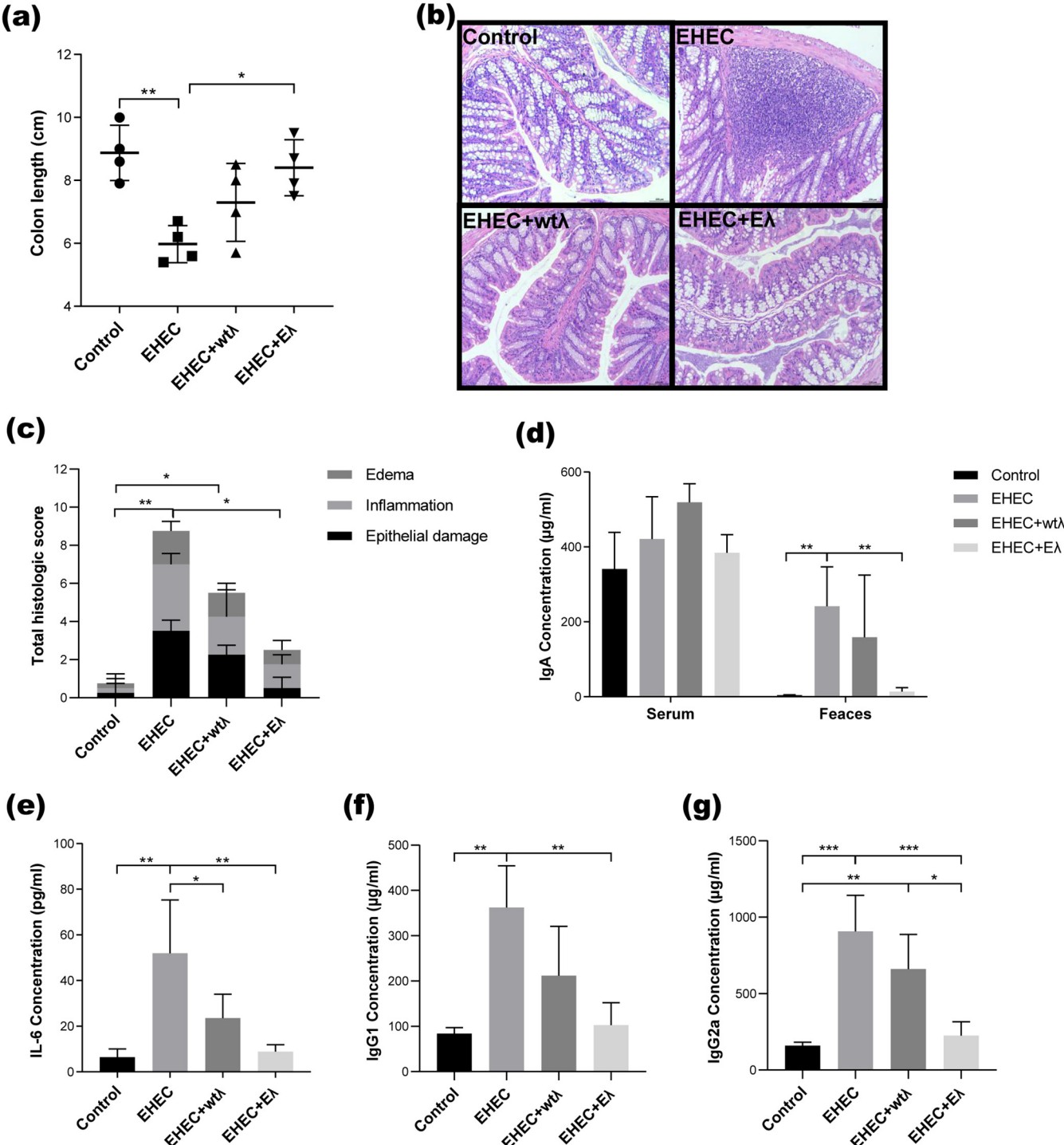

**FIG 3** The Eλ phage alleviated EHEC-induced tissue damage and inflammation markers. (a) Statistical analysis of colon lengths of all mice from the four groups. (b) Representative images of hematoxylin and eosin (H&E)-stained colon; the colon was taken from mice on day 4. Magnification, ×200; scale bars, 100 μm. (c) Histological scores summarizing the tissue damage from the images. Four mice were analyzed from each of the four groups. Each of the three variables, namely, edema, inflammation, and epithelial damage, was scored on a scale of 0 to 4 according to the evaluation criteria published by Selle et al. (24). (d) Levels of IgA from the serum (left) and feces (right). (e to g) Levels of IL-6 (e), IgG1 (f), and IgG2a (g) from mouse serum. All data are expressed as the mean ± SD ($n = 4$; *, $P < 0.05$; **, $P < 0.01$; ***, $P < 0.001$; one-way ANOVA and Tukey's posttest).

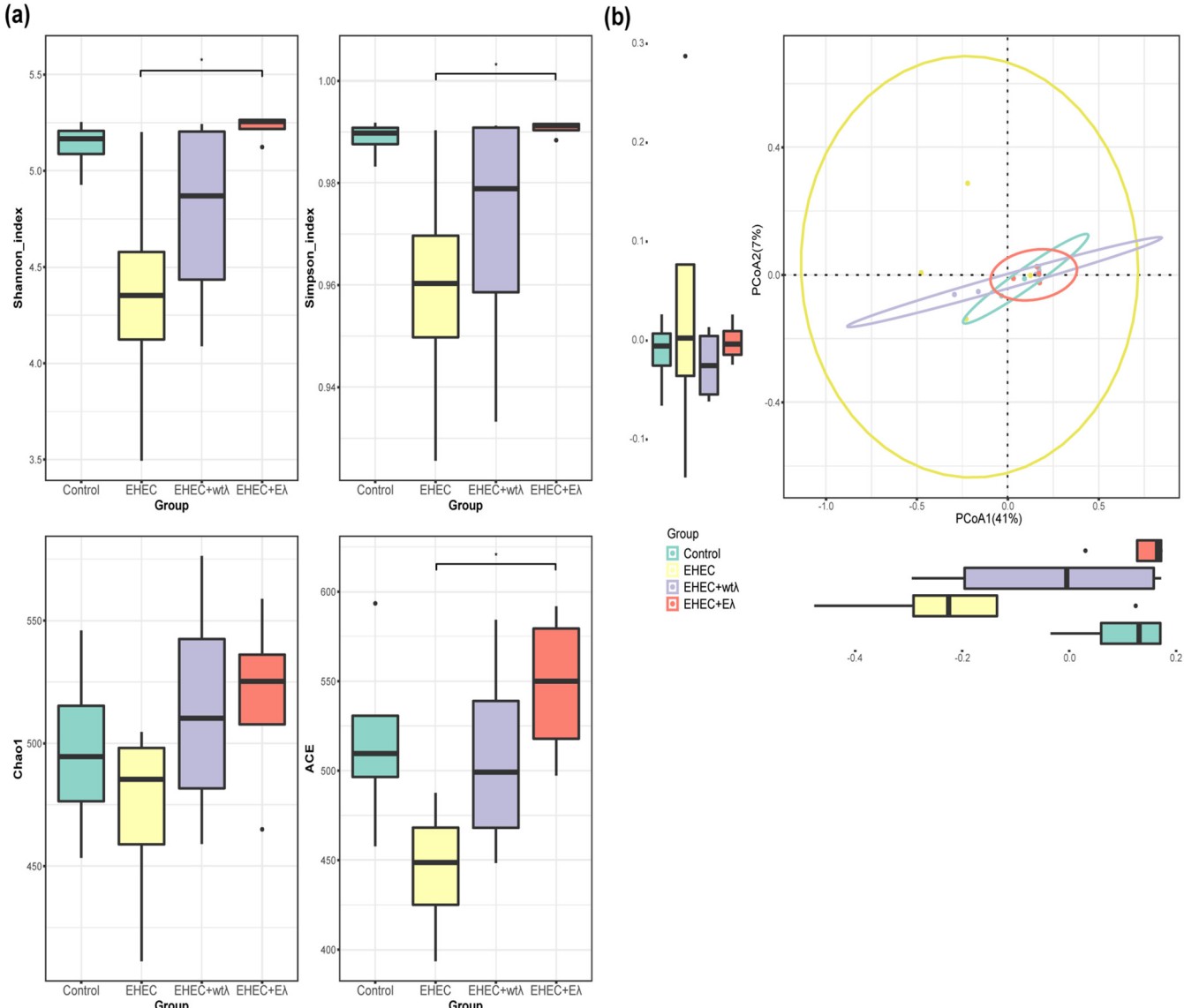

**FIG 4** The Eλ phage restored EHEC-induced gut microbiota dysbiosis. (a) Alpha diversity analysis of mouse gut microbiota community compositions of the four experimental groups measured using Shannon (left) and Simpson (right) indexes. (*n* = 4; *, *P* < 0.05; Wilcoxon rank-sum test). (b) Principal-coordinate analysis (PCoA) of the gut microbiota community compositions of the four groups (beta diversity).

dispersion, suggesting that EHEC infection indeed disrupted mouse gut microbiota, which could be effectively restored by Eλ treatment (Fig. 4b).

## DISCUSSION

Bacterial infections are responsible for many serious and fatal diseases (1). Entero-hemorrhagic *Escherichia coli* (EHEC) is a foodborne zoonotic infectious disease that has caused multiple outbreaks of bloody diarrhea and hemolytic uremic syndrome world-wide, posing a serious public health problem (35, 36). At present, the pathogenic mechanism and virulence factors of EHEC have been studied in depth, but the use of antibiotics in the treatment of EHEC infection is controversial. Studies have shown that due to the widespread use (or abuse) of antibiotics, the frequency of antibiotic-resistant bacteria is increasing gradually all over the world (37, 38). At the same time, the broad-spectrum bactericidal effect of antibiotics will directly affect the stability of the intestinal microbial community, thereby affecting the host intestinal immune system, directly or indirectly promoting the occurrence of diseases (39, 40). Previous studies

have demonstrated that phage preparations can reduce the viable count of pathogenic bacteria (41, 42). In addition, several studies have demonstrated that phage-mediated CRISPR-Cas systems can selectively target pathogenic bacteria in infection models (25, 43). In this study, we constructed an engineered phage carrying the CRISPR-Cas3 system and validated the engineered phage to kill EHEC with high efficiency and specificity *in vitro* and *in vivo*. To the best of our knowledge, this is the first report on engineered phage as a bactericide for EHEC.

Through *in vitro* experiments, we proved that the bactericidal effect of engineered λ phage (Eλ) was significantly better than that of the wild-type λ (wtλ). The wtλ could inhibit bacterial growth before fixed hours, but the bacterial growth reached the same level as the control group at the 17th hour. In contrast, Eλ completely inhibited bacterial growth throughout the experiment (18 h). At the same time, we found that at the MOI of 1, the EHEC treated by the Eλ resumed growth after ~8 h. Previous research has shown that under phage stress, bacteria can develop resistance to phages, rendering phage therapy useless (44, 45). In our study, recovered EHECs (recovered EHECs after 8 h, MOI of 1) were reinfected with Eλ to a final MOI of 10 to examine whether EHECs resumed growth due to resistance to Eλ. The results showed that Eλ could still effectively kill the recovered EHECs, suggesting that the growth recovery of EHECs is not due to the development of resistance to the Eλ but, rather, that the Eλ was added in insufficient amounts to allow EHECs to escape. Previous studies have shown that lytic phages isolated from animal sources could kill 99% of the EHECs (46), an efficiency comparable to our Eλ. However, host bacterial cells could quickly develop resistance to wild-type phage strains (47–49), supporting the advantage of engineering phages.

We also showed that the Eλ had high specificity against the EHEC strains. In this study, we tested the Eλ on multiple *E. coli* strains, and we observed a significant bactericidal effect on all the three EHEC strains, but no obvious effects on common laboratory and gut commensal *E. coli* strains. The specificity was achieved by targeting the *eae* gene. We noticed that although 97% of the 170 EHEC strains whose genomes were available in NCBI contained the *eae* gene, our Eλ might not target all the EHEC strains, especially those without the *eae* gene or whose *eae* genes did not have a perfect match with our CRISPR spacers. However, the targeted *E. coli* strains by our Eλ could be easily expanded by including more CRISPR spacers. In other words, the killing specificity of the Eλ is highly controllable.

The mouse EHEC infection models have been extensively used to study the pathogenesis of EHEC (50, 51). In a previous study, treatment of EHEC infection with a combined phage "cocktail" F.O.P. (an *E. coli*-, *Salmonella* spp.-, and *Listeria monocytogenes*-targeting bacteriophage cocktail) effectively reduced the intestinal EHEC load in mice (52). However, the changes in the gut microbial diversity of mice with combined phage treatment were highly similar to those of antibiotics, reflecting the same broad-spectrum bactericidal effect of combined phages as antibiotics, which could affect the growth of other strains while eliminating EHEC. In addition, previous studies showed that effective reduction of EHEC load could only be achieved by multiple administrations of combined phage therapy during the treatment period (53). Our animal experiments showed that after mice were infected with EHEC and given only one phage treatment, the survival rate of mice in the Eλ-treated group was 100%, much higher than 60% in the wtλ phage-treated group and 0% in the no-intervention group. The results of mouse fecal colony count showed that EHEC was hardly detected in the feces of the mice in the Eλ treatment group, indicating its high bactericidal efficiency.

Previous studies have shown that infection of mice with EHEC induces colonic damage (colon epithelial necrosis, neutrophilic colitis), which can subsequently progress from colitis to renal failure and secondary damage to internal organs such as the liver (54–57). Our study shows that Eλ can improve EHEC-induced colonic epithelial cell and mucosal lesions and inflammatory cell infiltration, as well as the pathological changes in kidney and liver cells.

In addition, the entry of EHEC into the human body can act on the intestinal tract and cause infection. The Shiga toxin secreted by EHEC easily causes severe intestinal inflammation, thereby promoting the expression of proinflammatory factors such as IL-6; at the same time, the host produces intestinal mucosal immune responses and systemic immune responses, which increase the levels of IgG1, IgG2a, and IgA (58). In our study, we found that the serum proinflammatory factor IL-6 and antibodies IgG1, IgG2a, and IgA were significantly reduced in infected mice treated with E$\lambda$ to levels comparable to the control. We again showed significantly better results in the E$\lambda$ group than in the wt$\lambda$ group.

Gut microbes have a symbiotic relationship with the host, and the two influence each other to maintain the homeostasis of the intestinal environment. Once this balance is broken, gut microbes will directly or indirectly promote the occurrence of diseases (39, 40). Previous studies have demonstrated that the use of phages to treat bacterial infections causes less disturbance to the gut microbiome (52). Our microbiota analysis demonstrated that while E$\lambda$ and wt$\lambda$ phages were similarly effective at reducing the levels of their targeted EHEC strain in mice, the E$\lambda$ was noticeably better in maintaining the natural richness and diversity of the gut commensal flora than the wt$\lambda$ phage. One pitfall of our mouse experiment was that only a limited number of mice ($n = 4$) were used in each group. However, even with the small group size, we still observed statistically significant differences in the microbial richness and diversities between the EHEC and E$\lambda$ groups, which strongly support our conclusion that the E$\lambda$ could better protect mice against the EHEC infection. Nevertheless, further validation using a larger number of mice per group will be needed in the future.

**Conclusions.** In this study, we engineered a lambda phage to efficiently and specifically target and eliminate EHEC, providing a new strategy for the treatment and prevention of EHEC infection. Both *in vitro* and *in vivo* experiments confirmed that the E$\lambda$ phage could completely suppress the growth of EHEC over an extended period of time. Our *in vivo* experiments further suggested the E$\lambda$ could restore EHEC-induced gut dysbiosis and support *in vivo* application of the engineered phages. In addition, unlike wild-type phages, our E$\lambda$ has the potential to overcome phage resistance by its target bacterium. We believe that our methods can be used to target other genes that are responsible for antibiotic resistance genes and/or human toxins and engineer other phages.

## MATERIALS AND METHODS

**Bacterial strains and culture conditions.** The bacterial strains and plasmids used in this study are listed in Table S2 in the supplemental material. Bacteria were grown in Luria-Bertani (LB) medium unless otherwise indicated.

**The knockout of the *cro* gene of lambda phage.** Lambda engineering was carried out in the *Escherichia coli* AB2013329 strain in which the lambda phage is incorporated as a prophage. The *cro* gene of lambda phage was knocked out using CRISPR-Cas9 gene editing technology (59). Briefly, single guide RNA (sgRNA) sequences targeting the *cro* gene were designed by the CRISPR tool (http://chopchop.cbu.uib.no/) and introduced into the pTargetF plasmid using the primers sgcroF and sgcroR by whole-plasmid PCR to generate plasmid pTargetF-sgcro (Table S2 and Table S3). The upstream homologous arm (742 bp) and downstream homologous arm (735 bp) of the flanking region of the target sites were amplified from the genome (wt$\lambda$ phage) by primer pairs Hom_cro_UP_F and Hom_cro_UP_R as well as Hom_cro_DW_F and Hom_cro_DW_R and infused by overlap PCR to generate donor DNA for knockouts (Donor-cro) (Tables S3 and S4).

To prepare competent cells, an overnight culture of *E. coli* AB2013329 (Table S2) was diluted 1:100 in 10 mL fresh LB at 37°C until the optical density at 600 nm (OD$_{600}$) reached $\sim$0.4. The culture was then centrifuged at 5,000 $\times$ *g* for 10 min at 4°C. The supernatant was removed, and the pellet was washed three times in ice-cold double distilled water (ddH$_2$O). We resuspended the pellet in 200 $\mu$L of ice-cold ddH$_2$O and kept it on ice to obtain *E. coli* AB2013329-competent cells. The pCas plasmid was electroporated into *E. coli* AB2013329-competent cells to obtain the *E. coli* AB2013329-pCas strain. One milliliter of an overnight culture of *E. coli* AB2013329-pCas was added into 100 mL fresh LB medium supplemented with kanamycin (50 $\mu$g/mL) and incubated at 30°C. Arabinose (10 mM) was added into the medium when the OD$_{600}$ of the culture reached 0.2, and the competent cells were prepared when the OD$_{600}$ of the culture reached at $\sim$0.4 to 0.6 as previously described. The pTargetF-cro and donor-cro were then electroporated into the competent, and the positive clones were screened by the primer pair ck_croF and ck_croR. After curing the plasmid pTargetF, the resulting strain was named AB2013329-$\lambda\Delta$cro.

**Plasmid curing.** To cure the plasmids derived from the pTargetF, the positive clones were incubated into 2 mL LB medium containing kanamycin (50 $\mu$g/mL) and IPTG (isopropyl-$\beta$-D-thiogalactopyranoside) (0.5 mmol/L) at 30°C for 12 h. Then, the culture was diluted and then plated on LB solid medium containing kanamycin (50 $\mu$g/mL) and incubated at 30°C. The colonies that showed sensitivity to spectinomycin were confirmed as cured. For the curing of pCas, the culture was incubated at 37°C overnight, and the clones sensitive to kanamycin were confirmed as cured.

**Integration of the *cas3* gene.** To insert the *cas3* gene required for CRISPR interference under the tac promoter, the Tac-Cas3 fragment containing the intact tac promoter and the front segment sequence of *cas3* was synthesized by GenScript (Nanjing, China) and ligated into the vector pMAL-c5x-TacCas3 (Table S2). Then, the Tac-Cas3 fragment was amplified by primers Tac_Cas3F and Tac_Cas3R using vector pMAL-c5x-TacCas3 as the template; meanwhile, the *cas3* gene was also amplified from *E. coli* MG1655. Then, the pTacCas3 fragment was generated by infusion of Tac-Cas3 and *cas3* via overlap PCR (Table S4). The region ~19014 to 27480 of the $\lambda$ genome was proven to have no effect on the structure and function of lambda phage (23) and was thus selected for the insertion site of the CRISPR-associated proteins. The sgRNA sequence targeting this region [SgRNA$_{lambda(Cas3)}$] was designed by the CRISPR tool (http://chopchop.cbu.uib.no/) and introduced into plasmid pTargetF using the primers sgCas3_1F/gCas3_1R and sgCas3_2F/sgCas3_2R by whole-plasmid PCR to generate plasmid pTargetF-sgRNA$_{lambda(Cas3)}$ (Tables S3 and S4). An overnight culture of *E. coli* AB2013329-$\lambda\Delta$cro (Table S2) harboring the pCas plasmid was diluted 1:100 in 50 mL fresh LB supplemented with kanamycin (50 $\mu$g/mL) and incubated at 30°C. Arabinose (10 mM) was added into the medium until the OD$_{600}$ reached ~0.2, and the competent cell was prepared when the OD$_{600}$ of the culture reached at ~0.4 to 0.6 as previously described. Then, pTargetF-sgRNA$_{lambda(Cas3)}$ and the PCR product containing pTacCas3 sequence and the flanking sequences of the insertion loci were introduced into the competent cells, and the colony with the correct insertion of Cas3pTac was identified by primers Hom_Cas3_UP_F and Hom_Cas3_DW_R. The positive strain was named *E. coli* AB2013329-$\lambda\Delta$cro::Cas3.

**Integration of *Cascade* genes, *Cas1* gene, and *Cas2* gene.** To insert *Cascade* genes (including the *CasA* gene, *CasB* gene, *CasC* gene, *CasD* gene, and *CasE* gene, or *CasABCDE*) required for CRISPR interference under tac promoter, the Tac-Cascade fragment containing the intact tac promoter and the front segment sequence of *CasA* was synthesized by GenScript and ligated into the vector pMAL-c5x-TacCacade (Table S2).

The Cascade fragment was amplified from the genome of MG1655 by primers T7_mg1655_CascadeF and T7_mg1655_CascadeR, and the pTac-Cascade fragment was amplified from the vector pMAL-c5x-TacCascade by primers Tac_CascadeF and Tac_CascadeR. The two fragments were then fused by overlap PCR to generate the pTacCascade fragment (Tables S3 and S4). Then, the fragment pTacCascade was introduced into the next gene locus of Cas3 by CRISPR-Cas9-mediated knock-in experiments as described before. Briefly, the sequence next to the inserted gene, *Cas3*, was selected for designing the sgRNA targets, and the designed sgRNA sequence [sgRNA$_{(Tac-Cascade)}$] was introduced into p-TargetF by whole-plasmid PCR to generate pTargetF-sgRNA$_{(Tac-Cascade)}$. Then, pTargetF-sgRNA$_{(Tac-Cascade)}$ and PCR products encoding the pTac-Cascade sequence and the flanking sequence of the insertion loci were introduced into the competent *E. coli* AB2013329-$\lambda\Delta$cro cell harboring plasmid pCas. Then, the colons with the correct insertion of pTac-Cascade were identified by primers Hom_Cascade_UP_F and Hom_Cascade_DW_R (Tables S3 and S4).

Then, the *cas12* (*cas1* gene and *cas2* gene) fragment was sequentially integrated into the genome of *E. coli* AB2013329 -$\lambda\Delta$cro::Cas3 in the same way as integration of Cascade, and the positive strain was named *E. coli* AB2013329-$\lambda\Delta$cro::Cas3::Cascade.

**Designing and integration of the CRISPR array.** To specifically target EHEC, the unique gene, *eae*, of EHEC was selected to design the self-target sgRNAs. A constitutive promoter was used to control the transcription of the CRISPR array. First, the fragment containing the intact J23119 (speI) promoter and the front sequence of CRISPR RNA (crRNA) was synthesized by GenScript and ligated into the plasmid pUC57 (Table S2). Then, the fragment was amplified by premiers pcrRNA_F and pcrRNA_R. Then, the CRISPR array containing 7 repeats and 6 intervening spacers was inserted into the genome locus 19014 to 27480 of the lambda phage by CRISPR-Cas9 system in a manner similar to that described above. The colony with correct insertion of crRNA was identified by primers ck_crRNAF and ck_crRNAR, and the positive strain was named *E. coli* AB2013329-$\lambda\Delta$cro::Cas3::Cascade::crRNA.

**Searching and downloading EHEC genomes from NCBI.** We searched the EHEC strains of O157:H7, O26:H11, O121:H19, and other serotypes from the NCBI prokaryotic genome database (https://www.ncbi.nlm.nih.gov/genome/browse#!/prokaryotes/) and downloaded a total of 170 genomes at the complete or chromosome levels using NCBI-genome-download (version 0.3.1; https://github.com/kblin/ncbi-genome-download/). Then, we downloaded the sequence of the *eae* gene (gene ID 915471) from NCBI from the gene database (https://www.ncbi.nlm.nih.gov/gene).

**Annotation of the *eae* gene in downloaded EHEC genomes.** We aligned the *eae* gene with the 170 EHEC strains genomes using nucleotide-nucleotide BLAST (default parameters) (version 2.9.0; 60) to identify whether the genome contains the *eae* gene. When the E value of alignment is less than 0.05, the genome is considered to contain the *eae* gene. Then, we aligned the target site (5′-ATGCTAACGGTAAGGCAACCGTAACGTTGAAGTCG-3′) with the 170 EHEC strain genomes using nucleotide-nucleotide BLAST (-task blastn-short -word_size 4) (version 2.9.0; 60), and we considered that the genomic had the target site only when the E value was less than 0.05.

The list of 170 EHEC genomes, their *eae* gene, and target site annotation results can be found in Table S1.

**Prophage induction. (i) Induction of WT lambda phage.** An overnight culture of *E. coli* AB2013329 was diluted 1:100 in 100 mL fresh LB at 37°C until the OD$_{600}$ reached 0.4 to 0.6. Five milliliters of the

culture were centrifuged at 5,000 × $g$ for 10 min and resuspended in the sediment with 1 mL sterilized water. The suspension was replaced in a new sterile dish without a lid and then exposed to a 365-nm UV lamp (0.15 mJ/cm$^2$) for 15 s. After irradiation, the suspension was transferred to 5 mL fresh LB and incubated at 37°C for 3 h. Then, 0.5 mL chloroform was added and shaken gently followed by centrifugation at 12,000 × $g$ for 10 min to remove cell or tissue debris. The supernatant was transferred to a new tube and diluted with PBS. Then, 100 $\mu$L of various dilutions were mixed with 100 $\mu$L/mL of $E.\ coli$ strain DH5$\alpha$ and used for double-layer agar plate assay and incubated at 37°C overnight.

**(ii) The induction of engineered lambda phage.** A single colony of $E.\ coli$ AB2013329-$\lambda\Delta$cro::Cas3::Cascade::crRNA host cells was inoculated in 3 mL LB medium and incubated at 37°C until the OD$_{600}$ reached ~0.5 followed by addition of 4× volume of fresh LB and mitomycin C (1.5 $\mu$g/mL). Then, the mixture was incubated at 37°C for 14 h. After incubation, 3 mL chloroform was added, and the mixture was then gently shaken for 15 min followed by centrifugation at 4,000 × $g$ for 30 min at 4°C. The mixture was then filtered with a 0.22-$\mu$m filter after the chloroform was evaporated.

**(iii) Killing efficiency of phages against EHEC.** One hundred microliters of EHEC (ATCC 35150) bacterial solution ($10^7$ CFU/mL) cultured overnight were inoculated into three tubes containing liquid LB at the ratio of 1:100. Then, 1% volume of wt$\lambda$ solution (100 $\mu$L, $10^8$ PFU/mL) and E$\lambda$ solution (100 $\mu$L, $10^8$ PFU/mL) were added in two tubes separately, and an equal volume of PBS was added to the remaining tube as a control. The tubes were incubated at 37°C for 18 h, and the OD$_{600}$ was measured every hour to determine the bacterial killing efficiency of phages.

**(iv) Killing specificity of phages against EHEC.** To test the killing specificity of wt$\lambda$ and E$\lambda$, in total, 22 $E.\ coli$ strains were used, including three EHEC strains (ATCC 35150, BMZ142226, and BMZ174482), 4 EPEC strains (BMZ146241, BMZ147484, BMZ146062, and BMZ146061), 5 common laboratory strains (BW25113, Nissle1917, MG1655, DH5$\alpha$, and BL21), and 10 gut commensal strains (C32E1, C29E1, C6E3, C38E2, C37E1, G30E1, G16E2, G3E1, G8E2, and G7E2) isolated from human feces in our laboratory. The six "BMZ" strains, i.e., BMZ142226, BMZ174482, BMZ146241, BMZ147484, BMZ146062, and BMZ146061; two EHEC; and four EPEC strains mentioned above were purchased from Mingzhoubio (Ningbo, China) in May 2022 (Table S2). To isolate the commensal $E.\ coli$ strains, feces from healthy donors were diluted and spread on the solid MacConkey medium, and the colonies with the morphology conforming to $E.\ coli$ were selected and verified using $E.\ coli$-specific PCR primers (27-F, 5′-AGAGTTTGATCCTGGCTCAG-3′; 1492-R, 5′-TACGACTTAACCCCAATCGC-3′). The PCR products were sequenced for further confirmation.

To identify the prevalence of the $eae$ gene in the above $E.\ coli$ strains, we performed PCR using $eae$ gene-specific primers (Eae-1F, 5′-TGTCGCACTAACAGTCGCTT-3′; Eae-1R, 5′-TGGTCAAGTTGTCGACCAGG-3′). Then, the sequencing results of the PCR product were compared with the sequence of the "target site" (5′-ATGCTAACGGTAAGGCAACCGTAACGTTGAAGTCG-3′) to confirm whether the strains contained the target site.

These were grown to an OD$_{600}$ of 0.5, and the wt$\lambda$ solution and E$\lambda$ solution were added to the culture at a ratio of 1:100. The equivalent volume of sterile PBS was added to the control tubes. Then, the mixture was incubated at 37°C, and viable cell count was assessed after 12 h by the serial dilution plate count method. For bacterial killing efficiency of phages, the efficiency was calculated using the following equation: bacterial survival rate = [phage CFU (12h)/PBS CFU (12 h)] × 100%.

**Animal studies.** Specific-pathogen-free (SPF)-grade male Kunming mice, weighing ~13 to 15 g, were purchased from the Experimental Animal Center of Hubei Province. All mice were acclimated to the experimental room for 2 days before treatment. Mice were randomly divided into 4 groups (10 per group), the control, EHEC, EHEC plus wt$\lambda$ treatment (wt$\lambda$), and the EHEC plus E$\lambda$ treatment (E$\lambda$) group. All mice were then made to fast (no food and water) for 12 h. After fasting, all mice were intraperitoneally injected with mitomycin (2.5 mg/kg of body weight), followed by administration of EHEC (ATCC 35150, 100 $\mu$L, $10^{10}$ CFU/mL) via gavage on day 1 for all but the control groups. All mice of the control group received an equal volume of PBS solution. The mice recovered to a normal chow diet after EHEC infection for 12 h. On day 2 (1 day after EHEC challenge), all mice were inoculated by gavage with 100 $\mu$L 10% (wt/vol) solution of sodium bicarbonate. Then, the wt$\lambda$ and E$\lambda$ groups were inoculated by gavage with 100 $\mu$L ($10^{10}$ PFU/mL) of wt$\lambda$ phage solution and 100 $\mu$L ($10^{10}$ PFU/mL) of E$\lambda$, respectively. The mice of the control and EHEC groups were gavage fed with an equal volume of PBS only. The weight of each mouse was recorded once a day, and the general health status of each mouse was observed twice a day. The overall experimental design is shown in Fig. 2a.

**Sample collection.** Fresh mouse feces were collected daily with sterile tweezers and equally assigned to 2 sterile cryopreservation tubes. One was immediately stored at −80°C, and the other was stored at 4°C for EHEC colony-forming experiments.

On day 4, 4 mice were randomly selected from each group, the eyeballs of the mice were removed, and blood was collected. The blood was centrifuged at 2,000 × $g$ for 10 min after storage at room temperature for 2 h. Mice were sacrificed by cervical dislocation; entire colons were immediately removed, and their lengths were measured. The colons, livers, and kidneys of sacrificed mice were removed and fixed in formalin (10%) for 48 h to prepare hematoxylin and eosin (H&E)-stained sections.

**Detection of EHEC loads in mouse feces.** The feces of EHEC-treated mice were washed with PBS three times and suspended with PBS (mass of feces/volume of PBS, 1:9). Then, the fecal suspension was diluted and plated on the solid MacConkey medium to calculate the load of EHEC in feces. Because MacConkey agar medium is not designed specifically for EHEC, to accurately calculate the quantity of EHEC, 50 to 100 colonies were selected from each plate, and then PCR was performed with EHEC-specific primers (TVEHEC-F, 5′-TTGCTGTGGATATACGAGGGC-3′; TVEHEC-R, 5′-TCCGTTGTCATGGAAACCG-3′). The

number of colonies of EHEC in the fecal suspension were then calculated as (number of colonies on the plate × positive rate of EHEC × dilution gradient)/(weight of fecal suspension) (Fig. 2d).

**Tissue morphology observation.** At the time of dissection, the colon, liver, and kidney of the mice were quickly fixed in 10% neutral formalin fixative. Then, the tissue was dehydrated, rendered transparent, and embedded with paraffin to prepare tissue sections. The sections were deparaffinized and stained with H&E sequentially. The tissue morphology was observed with an optical microscope (Eclipse, Nikon, Japan). Histological sections were coded, randomized, and scored in a blind manner by Servicebio Technology Co., Ltd. (Wuhan, China). Edema, inflammation (cellular infiltration), and epithelial damage for the cecum and colon were each scored on a scale of from 0 to 4, based on a previously published numerical scoring scheme (61).

**ELISA for assessment of antigen-specific antibodies and cytokines.** At the time of dissection, the detection of IgG1, IgG2a, IgA, and IL-6 from serum and IgA from feces were measured using enzyme-linked immunosorbent assay (ELISA) kits according to the manufacturer's instructions (Novus mouse IgA ELISA kit, Novus mouse IgG1 ELISA kit, Novus mouse IgG2a ELISA kit, and Thermo Fisher Scientific CN, IL-6 mouse uncoated ELISA kit).

**Intestinal microbiota analysis.** Samples of mouse fecal homogenates (Table S5) were delivered to Tsingke Biotechnology (Wuhan, China) where the 16S libraries were constructed and sequenced. Briefly, genomic DNA was extracted from the homogenates using the Power fecal DNA isolation kit DNA (Qiagen) according to the manufacturer's instructions. Genomic DNA of extracted was used to prepare amplicons via the enzymes of KOD FX Neo (Toyobo) and Phusion (NEB) following the manufacturer's instructions. 16S sequencing was performed on the V3 to V4 region of the 16S ribosomal DNA. The sequences of the forward and reverse primers were 5′-ACTCCTACGGGAGGCAGCA-3′ and 5′-GGACTAC HVGGGTWTCTAAT-3′, respectively. DNA libraries were multiplexed and loaded onto an Illumina NovaSeq 6000 PE250 instrument following the manufacturer's instructions (Illumina, San Diego, CA, USA), and paired-end sequencing with read length of 250 (PE250) was performed. The forward and reverse reads were truncated by cutting off the index and primer sequences and joined with at least 10-bp overlap. Quality filtering on joined sequences was performed, and sequences which did not fulfill the following criteria were discarded: sequence length of >50 bp, no ambiguous bases, mean quality score of ≥20. After quality filtering and purifying chimeric sequences, the resulting sequences were clustered into operational taxonomic units (OTUs) according to Silva (62) (release 128; http://www.arb-silva.de) databases (sequences similarity was set to 97%). Mothur (63) was used to calculate ACE, Chao1, Shannon, and Simpson estimators of alpha diversity. A principal-coordinate analysis (PCoA) of the beta diversity was also prepared using the QIIME2 (https://qiime2.org/) bioinformatics pipeline.

**Statistics and other bioinformatics analyses.** Log-rank (Mantel-Cox) test was conducted using GraphPad Prism to estimate the survival rate of mice. Representation of the $P$ values was *, $P < 0.05$, and ****, $P < 0.0001$. Wilcoxon rank-sum test was conducted using R (v4.0.5) to determine the significance of changes in the alpha diversity of the bacterial microbiomes from mouse fecal homogenates. Representation of the $P$ values was *, $P < 0.05$, and **, $P < 0.01$. Data analysis in addition to the above was performed using one-way ANOVA with GraphPad Prism. Data were shown as the mean ± SD, and $P$ values of <0.05 were considered statistically significant. Representation of the $P$ value was *, $P < 0.05$; **, $P < 0.01$; ***, $P < 0.001$; and ****, $P < 0.0001$.

**Data availability.** The raw 16S sequencing data used in this study are available in the China National Center for Bioinformation Genome Sequence Archive (CNCB GSA) database (64, 65) under the accession ID PRJCA008925.

## SUPPLEMENTAL MATERIAL

Supplemental material is available online only.
**SUPPLEMENTAL FILE 1**, PDF file, 4.5 MB.

## ACKNOWLEDGMENTS

W.-H.C. acknowledges support from the National Natural Science Fund (32070660), NNSF-VR Sino-Swedish Joint Research Program (82161138017), and National Key Research and Development Program of China (2019YFA0905600).

W.-H.C., Z.L., and H.W. designed and directed the research; M.J., J.C., and X.Z. participated in experiments; G.H. analyzed the mouse 16S data; and M.J. and J.C. wrote the paper with results from all authors. All authors read and approved the final manuscript.

We declare no competing interests.

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
