## [Reviewer comments · Microbiology Spectrum]

Microbiology Spectrum

An engineered λ phage enables enhanced and strain-specific killing of enterohemorrhagic *Escherichia coli*

Wei-Hua Chen, Zhi Liu, Hailei Wang, Menglu Jin, Jingchao Chen, Xueyang Zhao, and Guoru Hu

Corresponding Author(s): Wei-Hua Chen, Huazhong University of Science and Technology

Review Timeline:

Submission Date:	April 6, 2022
Editorial Decision:	April 28, 2022
Revision Received:	June 22, 2022
Accepted:	July 8, 2022

Editor: Jinxin Liu

Reviewer(s): Disclosure of reviewer identity is with reference to reviewer comments included in decision letter(s). The following individuals involved in review of your submission have agreed to reveal their identity: Jinhu Huang (Reviewer #2)

Transaction Report:

DOI: <https://doi.org/10.1128/spectrum.01271-22>

April 28, 2022

Prof. Wei-Hua Chen
Huazhong University of Science and Technology
Department of Bioinformatics and Systems Biology, College of Life Science and Technology
Luoyu Road 1037
Wuhan 430074
China

Re: Spectrum01271-22 (An engineered λ phage enables enhanced and strain-specific killing of enterohemorrhagic *Escherichia coli*)

Dear Prof. Wei-Hua Chen:

Link Not Available

Sincerely,

Jinxin Liu

Journals Department
Reviewer comments:

Reviewer #1 (Comments for the Author):

This is a well written manuscript reporting the genetic engineering and efficacy testing of a bacteriophage targeting EHEC. The authors provide sufficient details regarding their use of the CRISPR system and perform appropriate in vitro and in vivo experiments to test the efficacy. However, for the in vivo mouse experiemnts the authors fail to state the amount of challenge dose, they only provide the concentration. Also, comparing efficacy of the engineered lambda phage to that of w/t lambda doesn't allow for a comparison with other previously reported lytic phages targeting EHEC. The authors should discuss other recently published studies (Lukman, C., Yonathan, C., Magdalena, S. et al. Isolation and characterization of pathogenic

Escherichia coli bacteriophages from chicken and beef offal. BMC Res Notes 13, 8 (2020)) and compare levels or reported efficacy and specificity. Microbiota studies appear to suffer from low numbers (N=4/group?), affecting especially the PCoA analysis provided in Fig. 4B.

Reviewer #2 (Comments for the Author):

Jin and co-authors engineered a λ phage to target and eliminate Enterohemorrhagic Escherichia coli (EHEC) with enhanced efficiency and specificity, but did not develop resistance to the E λ and not affect the growth of gut microbiota. The work is encouraging and provides a new strategy for the treatment and prevention of EHEC infection.

Major:

1. Since not all EHEC strains contained the eae gene. It is better to briefly describe the association between them, and thus the E λ might be used to suppress partial EHEC strains (eae-positive) growth.
2. The killing specificity of E λ was performed against EHEC and additional 15 E. coli strains (non-EHEC in speculation). How about other EHEC strains? Other EHEC strains, both eae-positive and negative, should be included to make the conclusion. Further, enteropathogenic E. coli (EPEC), which also carry eae gene, can be added to the killing assays.

Minor:

-Fig. S1 (d), The size of DNA marker

-Line 169-170: "At the MOI of 1, the EHEC treated by the E λ resumed growth after ~eight hours (Fig. 1b)", should it be Fig. S3b?

-Line 176, "...including five lab strains and ten isolated from human feces". Please briefly explain the background.

-Line 255-256, "We found the lowest Chao1, ACE, Shannon and Simpson indexes in the EHEC group". What does these indexes stand for?

Staff Comments:

Preparing Revision Guidelines

Please return the manuscript within 60 days; if you cannot complete the modification within this time period, please contact me. If you do not wish to modify the manuscript and prefer to submit it to another journal, please notify me of your decision immediately so that the manuscript may be formally withdrawn from consideration by Microbiology Spectrum.

Resubmission of revised manuscript

‘Spectrum01271-22R1 (An engineered λ phage enables enhanced and strain-specific killing of enterohemorrhagic Escherichia coli)’

We are grateful to the reviewers for their valuable comments. We have revised the manuscript according to the comments and addressed their concerns point-by-point as follows:

Hyperlink:

Responses to reviewer #1

Responses to reviewer #2

Responses to reviewer #1:

Comments for the Author

This is a well written manuscript reporting the genetic engineering and efficacy testing of a bacteriophage targeting EHEC. The authors provide sufficient details regarding their use of the CRISPR system and perform appropriate in vitro and in vivo experiments to test the efficacy.

Response: We thank the reviewer for his/her positive evaluation of our research. We will address all the issues raised by the reviewer below.

Comments/questions #1:

For the in vivo mouse experiments the authors fail to state the amount of challenge dose, they only provide the concentration.

Response: We apology for the omission.

Actions: As suggested by the reviewer, we added the details of the gavage doses of EHEC (100 μ L, 10^{10} CFU/mL) and the phages (100 μ L, 10^{10} PFU/mL) to the revised manuscript, in lines #202-207. For reviewer's convenience, we also included the texts here: "All mice except those in the Control group were infected with 100 μ L EHEC (10^{10} CFU (colony forming units)/mL) by gavage, followed by gavage with 100 μ L of 10% sodium bicarbonate solution that could protect phage particles from damage by gastric acid. Then the mice received one of three different treatments: PBS buffer, 100 μ L wt λ (10^{10} PFU (plaque forming unit)/mL) or 100 μ L E λ (10^{10} PFU/mL) (Fig. 2a) according to their groups."

Comments/questions #2:

Also, comparing efficacy of the engineered lambda phage to that of w/t lambda doesn't allow for a comparison with other previously reported lytic phages targeting EHEC. The authors should discuss other recently published studies (Lukman, C., Yonathan, C., Magdalena, S. et al. Isolation and characterization of pathogenic Escherichia coli bacteriophages from chicken and beef offal. BMC Res Notes 13, 8 (2020)) and compare levels or reported efficacy and specificity.

Response: We thank the reviewer for the excellent suggestion. Indeed, results from this publication show that the lytic phages isolated from animal sources have very high killing efficiency of 99% against the EHECs, much higher than the wild-type λ and comparable with our engineered λ phage (E λ). The results are quite remarkable.

However, one advantage of our E λ over wildtype phages is the bacteriophage resistance. In the current study, we experimentally proved that bacteria did not develop resistance to E λ . Conversely, the EHEC cells quickly developed resistance to

the wild-type λ . Previous studies have demonstrated that bacteria could rapidly develop resistance to wild-type lytic phages (Ref. 48-50)

Action: We cited the publication recommended by the reviewer, compared the reported killing efficiency with ours, and revised manuscript accordingly. The revised texts could be found in lines #316-320, page 9. For reviewer's convenience, we also included the texts here: "Previous studies have shown that lytic phages isolated from animal sources could kill 99% of the EHECs (Ref. 47), an efficiency comparable to our E λ . However, host bacterial cells could quickly develop resistance to wildtype phage strains (Refs. 48-50), supporting the advantage of engineering phages."

Comments/questions #3:

Microbiota studies appear to suffer from low numbers (N=4/group?), affecting especially the PCoA analysis provided in Fig. 4B.

Response: We agree with the reviewer that the number of mice per group is limited in the microbiome studies. Since we already observed statistically significant differences in the ACE, Shannon and Simpson indices between the EHEC and E λ groups, which strongly support our conclusion that the E λ could better protect mice against the EHEC infection. We thus believe that the amount of data is sufficient to support our results. Nevertheless, we do agree with the reviewer that further validation using a larger number of samples will be needed in the future.

Action: We discussed the limitation of the small group size, and mentioned the required future work in the revised manuscript (in lines #372-378). For reviewer's convenience, we also included the texts here: "One pitfall of our mouse experiment was that only a limited number of mice (n=4) were used in each group. However, even with the small group size, we still observed statistically significant differences in the microbial richness and diversities between the EHEC and E λ groups, which strongly support our conclusion that the E λ could better protect mice against the EHEC infection. Nevertheless, further validation using a larger number of mice per group will be needed in the future."

Responses to reviewer #2:

Comments for the Author

Jin and co-authors engineered a λ phage to target and eliminate Enterohemorrhagic Escherichia coli (EHEC) with enhanced efficiency and specificity, but did not develop resistance to the $E\lambda$ and not affect the growth of gut microbiota. The work is encouraging and provides a new strategy for the treatment and prevention of EHEC infection.

Response: We thank the reviewer for his/her evaluation of our research. We will address all the issues raised by the reviewer below.

Major comments/questions #1:

*Since not all EHEC strains contained the *eae* gene. It is better to briefly describe the association between them, and thus the $E\lambda$ might be used to suppress partial EHEC strains (*eae*-positive) growth.*

Response: We agree with the reviewer. This is indeed an excellent and relevant question. **Actions:**

1. As the reviewer suggested, we conducted a systematic survey on the EHEC strains in the NCBI prokaryotic genome database (<https://www.ncbi.nlm.nih.gov/genome/browse#!/prokaryotes/>). We identified in total 170 EHEC strains with genome assembly levels of complete or chromosome, and supplemented Table S1 to sort out the prevalence of the *eae* genes and our CRISPR-spacers (target site). Among which, 165 (97%) contained the *eae* gene,

and 128 (74% out of the 170) had perfect matches with our CRISPR-spacers (target site). Thus, most of the EHEC strains could be targeted by the E λ .

2. We added a Fig. S1 to summarize the results, and brief descriptions of the results to the revised the manuscript in lines #151-154. For reviewer's convenience, we also included the texts here: “In fact, among the 170 EHEC strains we have surveyed in the NCBI Prokaryotic Genome Database (Table S1), 165 (97%) contained the *eae* gene (Methods; Fig. S1), suggesting that broad killing ability against the EHEC strains could be achieved by targeting the *eae* gene.”.
 3. We also briefly discuss our solutions to the *eae*-negative strains, and the *eae*-positive strains that do not contain the target site of our CRISPR-Cas3 system. See texts in lines #321-331 for details. For your convenience, we also included the texts here: “We noticed that although 97% of the 170 EHEC strains whose genomes were available in NCBI contained the *eae* gene, our E λ might not target all the EHEC strains, especially those without the *eae* gene, or whose *eae* genes did not have perfect match with our CRISPR-spacers. However, the targeted *E. coli* strains by our E λ could be easily expanded by including more CRISPR-spacers. In other words, the killing specificity of the E λ is highly controllable.”.
-

Major comments/questions #2:

The killing specificity of E λ was performed against EHEC and additional 15 E.

coli strains (non-EHEC in speculation). How about other EHEC strains ? Other EHEC strains, both eae-positive and negative, should be included to make the conclusion.

Response: We thank you for your suggestion.

Actions:

1. As you suggested, we purchased and tested two additional EHEC strains. They both contained the *eae* gene, and could be efficiently killed by E λ (revised Fig. 1c). Since our CRISPR-Cas3 system was designed to target specifically against the *eae* gene, *eae*-negative EHEC strains in principle would not be targeted and killed. However, our CRISPR-Cas3 system could be easily modified to including other spacers to target the *eae*-negative strains.
-
2. We revised the Fig. 1c and Fig. S4d to include the new experimental results.
 3. We also briefly discuss the limitation of our current CRISPR-Cas3 design, and possible solutions to *eae*-negative strains. See texts in lines #321-331 of the revised manuscript. For your convenience, we also included the texts here: “We noticed that although 97% of the 170 EHEC strains whose genomes were available in NCBI contained the *eae* gene, our E λ might not target all the EHEC strains, especially those without the *eae* gene, or whose *eae* genes did not have perfect match with our CRISPR-spacers. However, the targeted E. coli strains by our E λ could be easily expanded by including more CRISPR-spacers. In other words, the killing specificity of the E λ is highly controllable.”.
-

Major comments/questions #3:

Further, enteropathogenic E. coli (EPEC) , which also carry eae gene, can be added to the killing assays.

Actions:

1. As the reviewer suggested, we purchased and tested four EPEC strains. Two of the four strains could be infected by wildtype λ (Fig. S3), but only one carried the *eae* gene. As expected, the *eae*-positive EPEC strain could be effectively killed by our E λ (revised Fig. 1c).
2. We added a brief summary of these results to the revised manuscript in lines #192-195. For your convenience, we also included the texts here: “Interestingly, one out the four EPEC strains could also be effectively eliminated by the E λ , which contained the *eae* gene (the BMZ146241 strain, Fig. 1c; Table S2), suggesting the E λ could also be used to eliminate other *E. coli* strains containing the pathogenic *eae* gene.”.
3. We also added a brief discussion on the limitations and advantages of the CRISPR-Cas3 system in light of the new experimental results on the EHEC and EPEC strains. Overall, we believe that our current engineering strategy offer enhanced killing efficiency and “controllable” specificity. First, any *E. coli* strains that could be infected by the wildtype λ and contain the target sites specified by the CRISPR-spacers, could be efficiently killed (eliminated). Second, the CRISPR-spacers could be easily (re-)configured to specifically target intended strains.

See texts in lines #321-331 of the revised manuscript for details. For your convenience, we also included the texts here: “We noticed that although 97% of the 170 EHEC strains whose genomes were available in NCBI contained the *eae* gene, our E λ might not target all the EHEC strains, especially those without the *eae* gene, or whose *eae* genes did not have perfect match with our CRISPR-spacers. However, the targeted E. coli strains by our E λ could be easily expanded by including more CRISPR-spacers. In other words, the killing specificity of the E λ is highly controllable.”.

Minor comments #1:

- Fig. S1 (d), The size of DNA marker

Response: corrected.

Minor comments #2:

-Line 169-170: "At the MOI of 1, the EHEC treated by the E λ resumed growth after ~eight hours (Fig. 1b)", should it be Fig. S3b?

Response: We apology for the error. We have corrected this in the manuscript.

Minor comments #3:

-Line 176, "...including five lab strains and ten isolated from human feces". Please briefly explain the background.?

Response: Thank you for the suggestion.

Action: As suggested by the reviewer, we have supplemented the background introduction of ten isolated from human feces strains. See texts in lines #552-563 of the revised manuscript. For your convenience, we also included the texts here: “To isolate the commensal *E. coli* strains, feces from health donors were diluted and spread on the solid MacConkey medium, and the colonies with the morphology conforming to *E. coli* were selected, and verified using *E. coli*-specific PCR primers (27-F 5' -AGAGTTTGATCCTGGCTCAG- 3', 1492-R 5' -TACGACTTAACCCCAATCGC- 3'). The PCR products were sequenced for further confirmation.”.

Minor comments #4:

-Line 255-256, "We found the lowest Chao1, ACE, Shannon and Simpson indexes in the EHEC group". What does these indexes stand for?

Response: These indexes provide overall characteristics of the structure and composition of microbial communities.

Action: As suggested by the reviewers, we added brief descriptions of these indexes in lines #266-269 in the revised manuscript, and added two references (Refs. 34, 35). For your convenience, we also included the texts here: “We compared the overall microbial community structures between groups measured by richness (Chao1 and ACE) and diversity (Shannon, Simpson) (Refs. 34, 35), and observed lowest Chao1,

ACE, Shannon and Simpson indexes (alpha diversity measurements) in the EHEC group.”

July 8, 2022

Prof. Wei-Hua Chen
Huazhong University of Science and Technology
Department of Bioinformatics and Systems Biology, College of Life Science and Technology
Luoyu Road 1037
Wuhan 430074
China

Re: Spectrum01271-22R1 (An engineered λ phage enables enhanced and strain-specific killing of enterohemorrhagic *Escherichia coli*)

Dear Prof. Wei-Hua Chen:

Your manuscript has been accepted, and I am forwarding it to the ASM Journals Department for publication. You will be notified when your proofs are ready to be viewed.

Sincerely,

Jinxin Liu
Editor, Microbiology Spectrum
